# Resting-state EEG dynamics help explain differences in response control in ADHD: Insight into electrophysiological mechanisms and sex differences

Jonah Kember[1,2]*, Lauren Stepien[2], Erin Panda[2], Ayda Tekok-Kilic[2]

**1** Department of Neurology and Neurosurgery, McGill University, Montréal, Québec, Canada, **2** Department of Child and Youth Studies, Brock University, St. Catharine's, Ontario, Canada

* jonah.kember@mail.mcgill.ca

**Data Availability Statement:** Data may be acquired from the Child Mind Institute (http://fcon_1000.projects.nitrc.org/indi/cmi_healthy_brain_

## Abstract

Reductions in response control (greater reaction time variability and commission error rate) are consistently observed in those diagnosed with attention-deficit/hyperactivity disorder (ADHD). Previous research suggests these reductions arise from a dysregulation of large-scale cortical networks. Here, we extended our understanding of this cortical-network/response-control pathway important to the neurobiology of ADHD. First, we assessed how dynamic changes in three resting-state EEG network properties thought to be relevant to ADHD (phase-synchronization, modularity, oscillatory power) related with response control during a simple perceptual decision-making task in 112 children/adolescents (aged 8–16) with and without ADHD. Second, we tested whether these associations differed in males and females who were matched in age, ADHD-status and ADHD- subtype. We found that changes in oscillatory power (as opposed to phase-synchrony and modularity) are most related with response control, and that this relationship is stronger in ADHD compared to controls. Specifically, a tendency to dwell in an electrophysiological state characterized by high alpha/beta power (8-12/13-30Hz) and low delta/theta power (1-3/4-7Hz) supported response control, particularly in those with ADHD. Time in this state might reflect an increased initiation of alpha-suppression mechanisms, recruited by those with ADHD to suppress processing unfavourable to response control. We also found marginally significant evidence that this relationship is stronger in males compared to females, suggesting a distinct etiology for response control in the female presentation of ADHD.

## 1. Introduction

Reduced response control (higher reaction-time variability/commission error rate) is consistently reported in those with attention-deficit/hyperactivity disorder (ADHD) across a wide range of neuropsychological tasks, including those requiring relatively simple perceptual decisions [1–4]. This task-invariant reduction in response control, particularly increased reaction-time variability, is thought to be a core cognitive phenotype of the disorder, with moderate-to-large meta-analytic effect sizes existing between ADHD and typically developing children

network/) following the acquisition of a Data Usage Agreement.

**Funding:** The study was partially supported by Brock University Council for Research in Social Sciences (CRISS), awarded to ATK and EP. The funders had no role in study design, data collection and analysis, decision to publish, or preparation of the manuscript. There was no additional external funding received for this study.

**Competing interests:** The authors have declared that no competing interests exist.

(Hedge's $g$ = .76 across 319 studies) [5, 6]. Some theories posit that reduced response control results from a dysregulation of large-scale cortical networks–functional networks in the cortex that arise through transient changes in the coupling/decoupling of oscillatory neural activity [7–11]. Indeed, in resting-state paradigms, dwell times in functional networks that are less modular/more-integrated are associated with decreases in response control in both typically developing participants and those with ADHD [8, 12]. Mechanistically, this network dysregulation is thought to attenuate somatosensory processing, thereby limiting the capacity to accumulate and process decision-relevant sensory information from the environment (referred to as evidence accumulation) [4, 13, 14].

A leading candidate for the dysregulation that reduces response control is a lack of segregation (i.e., anticorrelated functional connectivity) between the default mode network (DMN) with salience (SAL) and dorsal attention (DAN) networks [9, 15–17]. Termed 'DMN interference', this reduced segregation is thought to arise from dysregulated phasic and low tonic dopamine signaling, and is particularly prominent when examining network dynamics (i.e., how functional connectivity/network topology changes over time) [18–21]. While the precise interactions that reduce response control are not yet clear (i.e., some evidence implicates dysregulated frontoparietal (FPN)–SAL interactions, as opposed to DMN interactions) [7], there is ample evidence that a dysregulation of the canonical large-scale cortical networks reduces response control, thus representing a putative biological pathway of a prominent cognitive phenotype in ADHD that is relatively independent of subtype [6–10]. However, while dysregulation in network topology appears most likely, other candidate mechanisms include reduced overall network communication (indexed by phase synchronization) [22, 23] and altered signaling within local subregions of the network (indexed by oscillatory power) [24, 25].

Here, we extend our understanding of this cortical-network/response-control pathway in two important ways. First, we narrow in on the nature of dysregulation. We do so by observing the dynamics of three distinct electrophysiological phenomena (network topology, phase synchronization, and oscillatory power) present in cortical networks during a resting-state paradigm in people with and without ADHD and asking to what extent these phenomena are associated with measures of response-control. Second, we test whether this association differs between males and females, and if so, whether this difference depends on ADHD status. There has been a historical neglect of research into how the mechanisms underlying ADHD symptoms differ by sex [26] In some cases, these sex differences complicate prognosis during clinical practice, as similar behaviours arise through distinct mechanisms, and therefore need to be treated through distinct interventions [27]. With this in mind, we make use of a sample stratified by sex such that females and males are equally represented.

## 1.1 Electrophysiological correlates of cortical network dysregulation

Electrophysiological networks are regulated by various dissociable neural mechanisms. Certain mechanisms, mediated by neuromodulatory systems, regulate the transfer of information between cortical regions by increasing/decreasing the synchrony of their activity (phase-synchronization) [28–31]. Other mechanisms regulate network topology, creating functionally specialized subsystems (indexed using modularity) [32, 33]. Still yet, certain mechanisms, primarily mediated by oscillatory theta/alpha/beta-power, regulate functional network activity by increasing local synchronous oscillations in task-irrelevant regions, which, through an information-theoretic lens, directly suppresses task-irrelevant processing (oscillatory power) [34, 35].

Previous research investigating the cortical-network/response-control pathway in ADHD has predominantly relied on fMRI, which is limited in its ability to dissociate these

mechanisms due to its relatively poor temporal resolution and inability to directly measure neural activity [7, 8, 11] However, they are readily distinguished using electroencephalogram (EEG) by measuring: (1) the strength of phase-synchronization (phase-lag index [36]), (2) the topology of the resultant networks (modularity [36, 37]), and (3) oscillatory power [38].

While each of these three candidate mechanisms appears to be disrupted in ADHD, we hypothesized that network topology (indexed using modularity) would be most predictive of response control, based on our suspicion that DMN-interference is the underlying disruption. Moreover, we expected modularity to be more sensitive to measures of response control than phase-synchrony and power, given: (1) previous source-localized EEG research investigating this phenomenon in ADHD found network topology to be more sensitive than both global phase-synchrony and power [39], and (2) research in typically developing children aged 9–12 found that task-related changes in the modularity of large-scale EEG networks (across theta, alpha and beta bands during a sustained attention task) were associated with reaction-time variability [40].

To test our hypothesis, we first examined how three measures of cortical activity (phase-synchrony, modularity, and power) across the canonical EEG frequency-bands (delta, theta, alpha, beta) dynamically changed during the resting-state. Then, for each of these three measures, we characterized the resultant dynamics as trajectories through a state-space using Gaussian Hidden Markov Models (HMMs), similar to research by [21]. Crucially, this approach allowed us to reduce our high-dimensional data into a simplified set of electrophysiological profiles (states) without arbitrarily averaging across time-windows. Finally, for each of our three measures, we examined whether the tendency to dwell in certain regions of these state-spaces (i.e., exhibit more or less of one of the characteristic profiles identified by the model) were associated with response control on a simple perceptual decision-making task. To foreshadow results–we found that, contrary to our primary hypothesis, dynamic changes in the *power* of large-scale cortical networks, as opposed to changes in their modularity or phase-synchrony, were most associated with differences in response control.

## 1.2 Sex differences

We also extended our understanding of the cortical-network/response-control pathway by testing whether it differed in males and females, which we hypothesized it would. In support of this hypothesis, many of the biological systems mediating this relationship show differences in males and females. For example, in 8–12-year-olds with an ADHD diagnosis, sex differences exist in the strength of resting-state functional connectivity between medial prefrontal cortex and the striatum (circuits crucial for perceptual decision-making) [41]. Second, typically-developing children show sex differences in the dynamics of EEG resting-state cortical networks that are relevant to response control: males tend to spend more time in states with low activation of regions overlapping with the dorsal attention network (DAN, examined using simultaneous EEG-fMRI), which is anticorrelated with the DMN in typically developing children and directly supports response control [42–44]. Finally, females and males differentially respond to dopamine reuptake inhibitors (i.e., methylphenidate), which is known to regulate cortical networks (by suppressing DMN activity) and increase response control [6, 45, 46]. Thus, converging evidence suggests the relationship between large-scale cortical network dynamics and response control may depend on sex, which we investigate here. However, while various sex differences in the systems mediating response control appear to exist, the specific impact these differences may have on electrophysiology is unclear. This is especially true given the electrophysiological mechanisms underlying response control are themselves poorly understood, as discussed above. Because of this, we leave this hypothesis undirected.

## 2. Materials and methods

Data from the Child Mind Institute's Healthy Brain Network biobank (HBN) [47] were used for analyses. The HBN is an open-source data set containing a wide array of neuroimaging, cognitive and phenotype data that was collected from children with and without psychiatric diagnoses for the purpose of biomarker discovery. In the current study, we made use of the resting state EEG data, the performance data from a separate perceptual decision-making task, and phenotype data from a subset of participants (selection criteria described below) that were available at the time of writing (releases 1 to 6). Data access was approved by CMI/Healthy Brain Network and the study was approved by the Research Ethics Board at Brock University. All data were anonymized prior to us receiving them from the Healthy Brain Network.

### 2.1 Participants

We applied a set of stringent, pre-registered participant selection criteria to decrease the behavioural heterogeneity of our sample and to highlight sex differences. First, all participants had to: be between 8 and 18 years of age, have undergone EEG recording for the resting-state task, and have completed at least 2 of the 3 contrast-change detection task blocks (24 trials per block). For the control group, participants had to have no clinician diagnoses; for the ADHD group, participants had to have a primary diagnosis of ADHD and no additional diagnoses.

After application of these selection criteria, females with ADHD were the least well-represented (N = 28). ADHD-males, control-males, and control-females were then selected to have an equivalent sample size. When selecting ADHD-males, participants were matched on both age (by year) and subtype. When selecting controls, participants were matched on age (by year). While our selection criteria allowed for participants to be between 8 and 18 years of age, the range of the resultant sample was 8.03 to 16.83 (see section 3.1 for sample characteristics).

To assess the behavioural heterogeneity of the sample, two-way ANOVAs were conducted to examine whether self-reported internalizing and externalizing behaviours, ADHD-traits, and anxiety (see section 2.3 for further details on self-reported tests), differed by sex, diagnosis, or their interaction. Specifically, we analyzed standardized scores on three scales from the Child Behaviour Checklist (Attention Problems, Externalizing, Internalizing) [48], two scales from Conners 3rd Edition Self-Report (Hyperactivity/Impulsivity, Inattention) [49] and total scores from the Screen for Child Anxiety Related Emotional Disorders (SCARED) [50].

### 2.2 Response control

To assess response control, we examined performance on the Healthy Brain Network Biobank's contrast-change detection task [47, 51]. In this task, participants were visually presented with two overlaid ring-like patterns (one tilted 45° to the right, one tilted 45° to the left, otherwise identical) that flickered at a constant rate. At the onset of each trial, the contrast of each pattern (right and left) started at 50%, before one would gradually shift (over the course of 1000ms) to a contrast of 100%, while the other would gradually shift to a contrast of 0%. Participants were instructed to press the right button with their right hand when they detected the right pattern increasing in contrast, and the left button with their left hand when they detected the left pattern increasing in contrast. This paradigm indexes the ability of participants to encode sensory information, accumulate decision-relevant sensory information, and execute motor responses appropriately. It is described extensively elsewhere [47, 51].

Trials with no responses were excluded from analysis. Three measures of response control were taken: mean reaction-time (RT-Mean: mean response time across all correct trials), reaction-time variability (RT-Variability: standard deviation of response times across all correct trials), and task-performance (portion of completed trials with a correct response). To validate

these output measures, we conducted two-way ANOVAs to examine whether they differed by diagnosis (as expected), sex, or their interaction.

## 2.3 Questionnaires

**2.3.1 Child Behaviour Checklist (CBCL).** The CBCL assesses internalizing and externalizing behaviours that children and adolescents have experienced in the previous six months using 113 items on a 3-point likert scale (0 = Absent, 1 = Sometimes, 2 = Occurs often). Three subscales were used: Attention Problems (8 items), Externalizing behaviours (34 items), and Internalizing behaviours (33 items) [48]. These scales have high internal consistency, with the mean Cronbach's alpha at .90 [52].

**2.3.2 Conners 3rd Edition Self-Report/Short Form (C 3-SR-S).** C 3-SR-S was used to assess the cognitive and behavioural symptomology of ADHD [49]. C3-SR-2 has 39 items (0–3, *not true at all/never to very much true/very frequently*) measuring Inattention, Hyperactivity/Impulsivity, Learning Problems, Defiance/Aggression, Family Relations as well as 2 validity scales (Positive and Negative Impression. In this study, Hyperactivity/Impulsivity (5 items) and Inattention scales (6 items) were used. These scales have high internal consistency, with the mean Cronbach's alpha at .90 [49].

**2.3.3 Screen for Child Anxiety Related Emotional Disorders-Child (SCARED/C).** Total scores from SCARED were used to ensure anxiety (somatic/panic, general anxiety, separation anxiety, social phobia, school phobia) did not differ between Males/Females and ADHD/Controls [50]. SCARED is a 41 item checklist that uses a 3-point Likert-scale (0 = Not true/hardly ever true, 1 = Somewhat true/sometimes true, 2 = True/often true). These scales also have high internal consistency, with Cronbach's alpha ranging from .74 to .93 [50].

## 2.4 Resting-state EEG

**2.4.1 EEG recording and preprocessing.** High-density (128-channel) EEG was recorded using a Geodesic Hydrocel net at a sampling rate of 500 Hz during a resting-state paradigm. This paradigm involved alternating between 20 second periods of eyes-open (EO) and 40 second periods of eyes-closed (EC) rest (5 minutes total). EO and EC conditions both entered analyses. Resting-state EEG data had been previously preprocessed using an automated pipeline developed by [53]. In brief, this pipeline involved: (1) removing a set of 17 electrodes surrounding the chin and neck, (2) rejecting channels with variance > 3 SDs from the mean across all other channels, (3) applying a high-pass filter (0.1 Hz) and notch filter (59–61 Hz), and (4) correcting for ocular artifacts by regressing the eye channels from the scalp channels. From there, we also removed the channels used for ocular correction (E8, E14, E21, E25), leaving 107 scalp channels for analysis. A surface Laplacian spatial filter was then applied (recommended when analyzing phase-synchronization; implemented in MATLAB using code provided by [38]).

**2.4.2 EEG dynamics: Phase synchrony, modularity, and power.** To characterize dynamic changes in participants' cortical network dynamics during the resting-state paradigm, we calculated global measures (i.e., across all sensors) of frequency-band specific phase synchrony, modularity, and power using a sliding time-window approach. First, EEG data were *z*-scored, and bandpass finite impulse response filters were applied to the preprocessed data (Delta = 1-3Hz, Theta = 4-7Hz, Alpha = 8-12Hz, Beta = 13-30Hz; transitions = +/−.2 times the mean of each frequency-band). Filtering was done using the Signal Processing toolbox in MATLAB *R2021a* [54]. The order of the filters was estimated using the *kaiserord* function (passband ripple = 10%), filters were created using *fir1*, and filters were applied using *filtfilt*.

*To calculate phase-synchrony*, the instantaneous phase of the filtered signals was estimated using the Hilbert transform, and the sensor-by-sensor synchronization was calculated using the phase-lag index (PLI) [55]. PLI indexes how asymmetrically a set of phase-differences between signals *i* and *j* across a set of time-points are distributed; an asymmetric distribution of phase differences is said to have stronger phase synchrony. By calculating the *asymmetry* of the distribution, PLI attenuates zero-lag synchrony (and thus, presumably, synchrony that occurs as a result of single-source volume conduction).

*To calculate modularity*, an adjacency matrix ($A_{ij}$) was defined with our 107 EEG sensors as nodes and the sensor-sensor PLI values as edges. Then, Louvain's algorithm (implemented in the Brain Connectivity toolbox) [36] was used to find the partition of this network into sets of nodes that roughly maximized the modularity statistic:

$$Q = \frac{1}{v} \sum_{ij} (A_{ij} - e_{ij}) \, \delta(m_i, m_j)$$

Where δ is the Kronecker delta function (whose output is 1 when nodes *i* and *j* belong to the same module, *m*, and 0 otherwise), $e_{ij}$ is a null-model network with preserved connection weight but random topology, and *v* is the total network connectivity (sum of the upper triangular adjacency matrix). The average *Q* value across 100 runs of Louvain's algorithm was taken, and this process was repeated for all time-points and frequency-bands.

*To calculate power*, the real part of the Hilbert-transformed narrow-band signal taken previously to calculate phase was taken. This was squared to obtain an estimate of power, and the mean value of power within each time window was calculated [38].

The length of the sliding windows (i.e., number of time-points used to calculate PLI) were frequency-band specific, selected such that the temporal degrees of freedom (η) were matched across frequencies [56]. η is calculated as: $2B_wD$, where $B_w$ is the bandwidth (Hz), and *D* is the length of the sliding window (seconds). The uncertainty ($1/\sqrt{\eta}$) was kept at .15, corresponding to window lengths of: Delta = 11.11s, Theta = 7.40s, Alpha = 3.70s, Beta = 1.48s. To reduce computational demands, connectivity was calculated in 1 second increments. This provided each participant with a sequence of 300 observations (5 minutes of resting-state data), where each observation is a four-dimensional vector with frequency-band specific values of either: modularity, PLI, or power. All the above analyses were conducted in MATLAB R2021a [54].

**2.4.3 Hidden Markov models (HMMs): Characterizing dynamic changes in EEG metrics.** The sequences of observations provided by the above analyses can be thought of as trajectories through a four-dimensional space (defined separately for one of: PLI, modularity, or power across the four frequency-bands of interest: delta, theta, alpha, and beta). To characterize these trajectories, we first modeled them as Markovian processes emitted from unknown/hidden 'states' (Gaussian Hidden Markov models; HMMs). These models assume that observations are emitted from one of *n* multivariate Gaussian distributions (*n* specified *a priori*). Each Gaussian is referred to as a state and is described in its entirety by a mean vector (*μ*; denoting the mean of each dimension), a covariance matrix (*Σ*; denoting the variance of each dimension/covariance between dimensions) and an initial probability (across all states, initial probabilities are captured in the vector π). Transitions between states are described by a transition probability matrix (*P*; denotes the probability of transitioning from state *i* at time *t* to state *j* at time *t* + 1). Once states have been identified, observations can be classified into states using maximum likelihood estimation, and the portion of time spent in each state can be calculated (dwell-time). This modeling allowed us to reduce each participant's set of 300*4 observations into a set of *n* 'dwell-times' (dwell-time: the proportion of observations spent in state *n*).

HMM analyses were conducted using the *R* package *RcppHMM* [57]. Observations were *z*-score normalized and model parameters were randomly initialized using *initGHMM*. Parameters were estimated using the Baum-Welch expectation maximization algorithm (*learnEM*; convergence was reached when consecutive iterations differed by < .005). A set of 10 random initializations were used, and the model with the highest log-likelihood (i.e., that which explained the data the best) was taken as the final model. Following parameter estimation, sequences of observations were classified into states using maximum likelihood estimation implemented via the *Viterbi* algorithm. Dwell-times were then calculated for each state. All hidden Markov modeling was conducted in the programming language *R* [58].

We present results with the number of states (*n*) set to 4. However, to assess the sensitivity of our results to this free parameter, we explored analyses across a range of states (3 to 6; see section 3.4).

**2.4.4 Validation analyses: Testing for differences between EO and EC rest.** To increase our confidence that the resting-state EEG measures derived from the HMMs were indeed capturing meaningful physiological processes, we tested whether they differed between eyes-open rest (EO) and eyes-closed rest (EC). In fMRI, functional connectivity between default mode and salience networks, which is associated with reduced response control, is known to differ between EO and EC rest [8, 59]. Thus, we reasoned similar observations should be made here prior to testing our hypotheses. As such, we tested whether the dwell-times of each state significantly differed between EO and EC rest using independent samples t-tests (or, when variances were unequal, Welch's t-tests).

**2.4.5 Oscillatory dynamics, phenotype measures and sex differences.** We tested whether dwell-times within each state (which reflect a tendency to exhibit a particular electrophysiological profile) differed between males/females and ADHD/Controls using two-way ANOVAs. We then tested if they could predict our phenotype measures (self-report ADHD-related behaviours, age) using ordinary least squares regression (OLS).

## 2.5 Primary hypothesis

Our primary hypothesis was that there would be a significant association between response control and cortex-wide oscillatory dynamics. Specifically, we predicted that dwell time in a less modular network topology would relate to reduced response control, particularly in those with ADHD. To test this, we conducted OLS regression to predict response control from dwell-times (narrowing in on the response control measures that specifically differed between ADHD and controls). Please note that due to the structural multicollinearity that results from the way dwell-times are defined (i.e., in a 2-state situation, a participant with a dwell-time of .75 in state 1 necessarily has a dwell-time of .25 in state 2) there are 1 fewer degrees of freedom in all models than one might otherwise expect given the number of states.

## 2.6 Secondary hypothesis

Our secondary hypothesis was that the relationship between response control and cortical network dynamics would be significantly moderated by sex. To test this hypothesis, we first added a dummy-coded variable to the OLS regression models used to test our primary hypotheses, to avoid interpreting any main effects of sex as interactions (Males = 0, Females = 1). Then, to test for moderating effects, we examined whether these models were significantly more predictive following the inclusion of interaction terms (dwell-times multiplied by our sex-coded variable). To do so, we used the *F*-test for the joint significance of a subset of variables, where the

subset of variables were the interaction terms. This test is calculated as follows:

$$F(p - q, n - p - 1) = \frac{(SSE_R - SSE_F)/(p - q)}{MSE_F}$$

Where $SSE_R$ is the sum-of-squared errors of the reduced model (i.e., the model with dwell-times and diagnosis), $SSE_F$ is the sum-of-squared errors of the full model (i.e., with dwell-times, diagnosis, and dwell-times*diagnosis interactions), $MSE_F$ is the mean-squared error of the full model, $p$ is the number of features in the full model, and $q$ is the number of interaction terms. The numerator of this test calculates the average reduction in squared error per interaction term; the denominator normalizes this value by model performance.

## 3. Results

### 3.1 Sample characteristics

A total of 112 participants (28 ADHD-Females, 28 ADHD-Males, 28 Control-Males, 28 Control-Females), with an average age of 11.62 years old (SD = 2.42; Range = 8.03 to 16.83), entered analysis. Of the 56 diagnosed with ADHD, 36 presented with the DSM-V Inattentive subtype, 18 with the DSM-V Combined subtype, and 2 with the DSM-V Hyperactive/Impulsive subtype (all subtypes matched between males and females).

Means and standard deviations for age, self-reported ADHD behaviours, self-reported anxiety levels, and response control (mean reaction-time, reaction-time variability, proportion of trials with a correct response) are presented in Table 1, and $p$-values from the two-way ANOVAs (with sex and diagnosis as factors) are presented in Table 2. As shown in Table 2, significant main effects of diagnosis were present on all but three measures: (1) age, which had been matched during participant selection (p = .25), (2) self-reported anxiety, which was selected *a priori* as a control measure (p = .45), and (3) self-reported hyperactivity/impulsivity (p = .07), which may reflect our greater representation of ADHD-I, as opposed to ADHD-C/AHDH-HI diagnoses. Notably, there were no significant sex differences or interactions between sex and diagnosis on any of the measures examined (age, self-reported behaviours, response control).

**Table 1. Descriptive statistics.**

|  | Female (N = 56) | | Male (N = 56) | |
| --- | --- | --- | --- | --- |
|  | ADHD (N = 28) | No Diagnosis (N = 28) | ADHD (N = 28) | No Diagnosis (N = 28) |
| **Age** | 11.74 (2.37) | 11.67 (2.55) | 12.03 (2.54) | 11.04 (2.23) |
| **CBCL- Attention problems** | 68.11 (12.73) | 55.39 (8.66) | 66.79 (11.22) | 53.36 (4.72) |
| **CBCL- Externalizing** | 58.32 (12.42) | 47.54 (10.33) | 56.68 (9.93) | 44.57 (11.27) |
| **CBCL- Internalizing** | 56.93 (12.81) | 48.14 (10.12) | 53.36 (8.65) | 48.21 (9.31) |
| **CSR- Hyperactivity/Impulsivity** | 62.04 (12.74) | 54.82 (12.14) | 55.89 (15.84) | 57.25 (10.59) |
| **CSR- Inattention** | 70.46 (16.18) | 56.11 (13.96) | 60.62 (19.14) | 58.07 (12.55) |
| **SCARED (Anxiety)** | 18.64 (11.73) | 15.32 (9.91) | 16.47 (13.1) | 18.29 (13.06) |
| **RT mean (seconds)** | 1.64 (0.19) | 1.65 (0.19) | 1.64 (0.27) | 1.59 (0.35) |
| **RT SD (seconds)** | 0.38 (0.15) | 0.32 (0.09) | 0.37 (0.08) | 0.32 (0.10) |
| **Task performance (proportion of correct responses)** | 0.72 (0.26) | 0.86 (0.18) | 0.76 (0.20) | 0.82 (0.25) |

Means are reported with standard deviation in brackets. Abbreviations are as follows: RT: reaction time, SD: standard deviation, CBCL: Child behaviour checklist, CSR: Conner's self report, SCARED: Screen for child anxiety related disorders.

**Table 2. Two-way ANOVA results.**

| | p-values | | |
|---|---|---|---|
| | **Sex** | **Diagnosis** | **Sex * Diagnosis** |
| Age | 0.72 | 0.25 | 0.33 |
| CBCL- Attention problems | 0.38 | 8.2 x 1011 ** | 0.79 |
| CBCL- Externalizing | 0.27 | 2.7 x 10–7 ** | 0.75 |
| CBCL- Internalizing | 0.40 | 5.6 x 10–4 ** | 0.38 |
| CSR- Hyperactivity/Impulsivity | 0.72 | 7.0 x 10–4 ** | 0.15 |
| CSR- Inattention | 0.32 | 7.0 x 10–2 | 8.2 x 10–2 * |
| SCARED (Anxiety) | 0.99 | 0.45 | 0.31 |
| RT mean | 0.99 | 0.77 | 0.81 |
| RT SD | 0.90 | 1.0 x 10–2 * | 0.82 |
| Task performance | 0.71 | 4.7 x 10–3 * | 0.54 |

* p < .05,

** p << .05.

Presented are results from the two-way ANOVAs comparing metrics of interest between males/females (sex) and ADHD/controls (diagnosis). Abbreviations are as follows: RT: reaction time, SD: standard deviation, CBCL: Child behaviour checklist, CSR: Conner's self-report, SCARED: Screen for child anxiety related disorders.

## 3.2 Validation analyses

**3.2.1 Response control.** Across all participants, an average of 67.39 ($SD$ = 10.07) trials entered analyses. This did not significantly differ by ADHD-diagnosis or sex ($p's \geq$ .68). Response control did in fact differ between those with and without an ADHD diagnosis: two-way ANOVAs revealed a main effect of diagnosis on reaction-time variability (F(1, 109) = 6.90, $p$ = .010; lower in controls compared to ADHD) and task performance (F(1, 109) = 8.38, $p$ = .005; higher in controls compared to ADHD), but not mean reaction-time ($p$ = .77). Box-plots illustrating these effects are presented in Fig 1. There were no main effects of sex, and no interactions between sex and diagnosis, on any of our response control measures (see Table 2).

**3.2.2 Hidden Markov modeling.** The $\mu$ parameters (means of the multivariate Gaussians) estimated for our three hidden Markov models (HMM-Modularity, HMM-PLI and

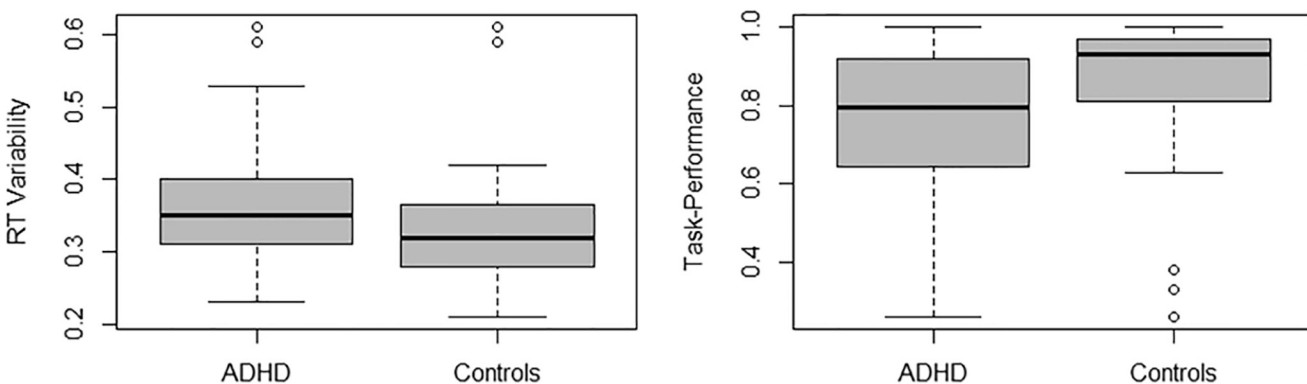

**Fig 1. Response control differed between ADHD and controls.** Boxplots showing differences in reaction-time variability and task performance in ADHD and controls.

HMM-Power) are presented in Table 3 along with dwell-times. For validation, we examined whether dwell-times in each state significantly differed between eyes-open (EO)and eyes-closed (EC) rest using Welch's unequal variances $t$-test. While dwell-times in the majority of states identified by our HMM-Power model differed significantly between EO and EC rest ($p$-values: S1: 6,0e-2, S2: 2.2e-16, S3: 2.0e-2, S4: 1.0e-9), dwell-times acquired from our HMM-Modularity and HMM-PLI models did not ($p$'s > .30). Thus, the cortex-wide dynamics captured by oscillatory power, as opposed to those captured by modularity or PLI, appear to have the highest validity (as we previously defined in section 2.4.4). The dynamics captured by the HMM-Power model (the proportion of participants in each state throughout the resting-state period) are shown in Fig 2A, and the specific electrophysiological profiles captured by each state are shown in Fig 2B.

To understand whether the states identified by each of these models (HMM-Modularity, HMM-PLI, HMM-Power) were sensitive to individual differences in behaviour, we tested whether dwell-times were able to predict age and our five self-report ADHD-related measures (CBCL-Attention problems, CBCL-Externalizing, CBCL-Internalizing, C3SR-Inattention, C3SR-Hyperactivity/Impulsivity). Using our HMM-Modularity model, dwell-times were unable to predict any measures. Using our HMM-PLI model, dwell-times were able to predict age ($F(3, 108) = 3.72$, $R^2 = .094$, $p = .014$), but no other measures. Using our HMM-Power model, dwell-times were able to predict age ($F(3, 108) = 2.78$, $R^2 = .071$, $p = .045$), CBCL-Attention Problems ($F(3, 108) = 2.79$, $R^2 = .072$, $p = .044$), and, at a marginally significant level, CS3R-Inattention ($F(3, 108) = 2.61$, $R^2 = .068$, $p = .055$). Thus, the dynamic changes in oscillatory power (as opposed to phase-synchrony or network modularity) not only appear to be the most valid (best distinguish EO from EC rest) but have the greatest sensitivity to individual differences in age and attention problems.

Inspection of the bivariate correlations that HMM-power dwell-times showed with age and attention problems clarified the effects: age was negatively correlated with dwell-time in state 3 ($r = -.25$, $p = .010$), and inattention was positively correlated with dwell-time in state 1 ($r = .22$, $p = .019$). Scatterplots for each effect are presented in Fig 3.

**Table 3. Hidden Markov $\mu$ parameters.**

| | | HMM-$\mu$ | | | | Dwell-Time |
|---|---|---|---|---|---|---|
| | | Delta | Theta | Alpha | Beta | |
| **Modularity** | State 1 | -0.44 | -0.26 | -0.16 | -0.17 | .31 (.31) |
| | State 2 | -1.12 | -1.10 | -1.01 | -0.97 | .19 (.35) |
| | State 3 | 0.45 | 0.36 | 0.34 | .30 | .39 (.34) |
| | State 4 | 1.46 | 1.25 | .92 | 1.05 | .12 (.25) |
| **PLI** | State 1 | 0.55 | 0.41 | -0.26 | 0.01 | .36 (.21) |
| | State 2 | -0.18 | -0.12 | 1.44 | 0.07 | .17 (.41) |
| | State 3 | -0.59 | -0.43 | -0.40 | -0.22 | .43 (.25) |
| | State 4 | 1.99 | 1.33 | 0.07 | 1.70 | .04 (.06) |
| **Power** | State 1 | 1.50 | 1.25 | 0.35 | 0.83 | .14 (.03) |
| | State 2 | -0.64 | -0.39 | 0.61 | 0.13 | .28 (.10) |
| | State 3 | 0.31 | 0.26 | -0.16 | -0.13 | .29 (.11) |
| | State 4 | -0.43 | -0.49 | -0.58 | -0.40 | .30 (.17) |

Presented are the $\mu$ parameters that define each state for modularity, PLI and power, as well as the mean dwell-time in each state across participants (with the standard deviation in brackets).

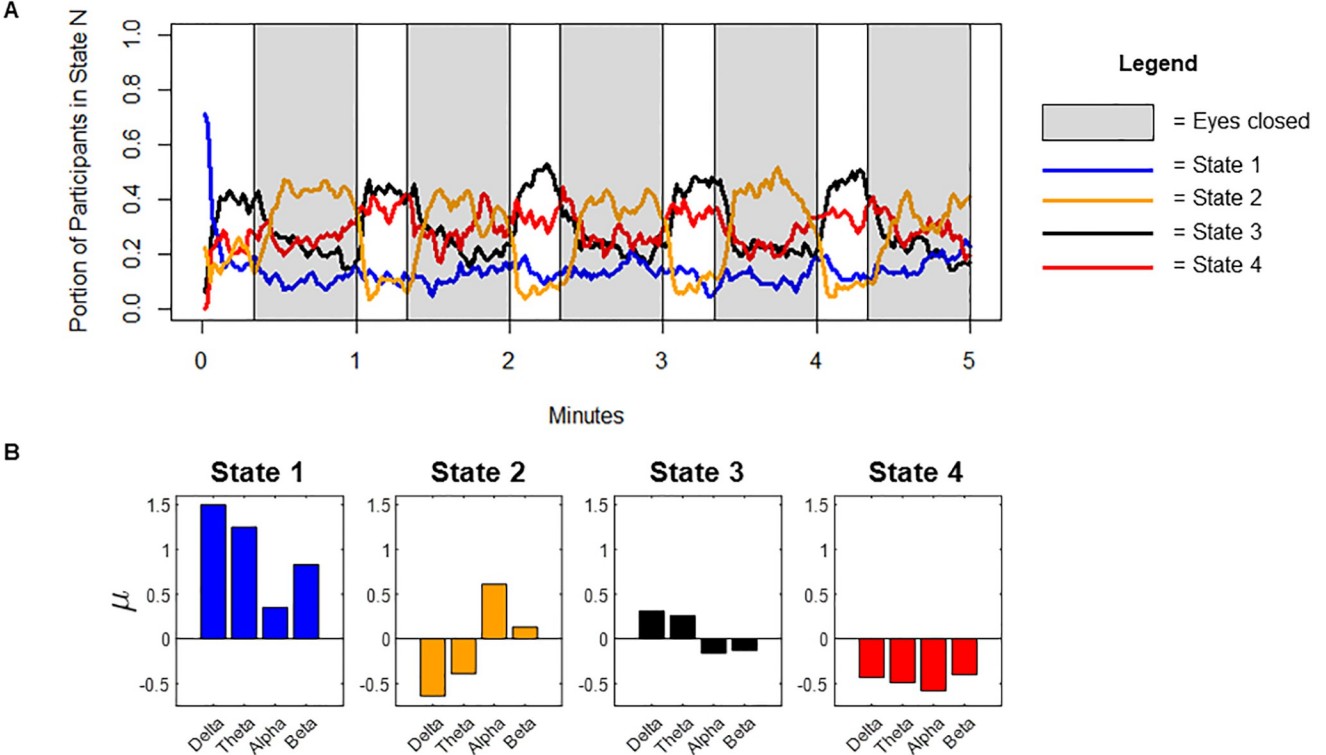

**Fig 2. Cortex-wide dynamics identified by HMM-power model.** (A) Portion of participants in each state at a given time-point throughout the resting state paradigm. (B) $\mu$ coefficients from the HMM-power model, showing the specific electrophysiological profile of each state.

### 3.3 Hypothesis testing

**3.3.1 Primary hypothesis.** We focused our primary hypothesis on the two measures that differed between ADHD and controls: reaction-time variability and task performance. We did not find support for our primary hypothesis: dwell-times from our HMM-Modularity model

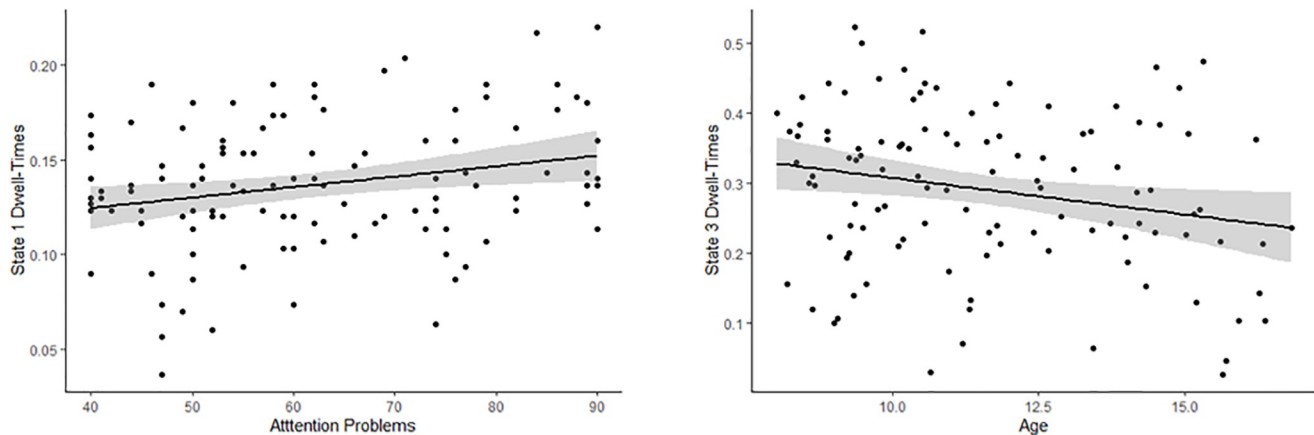

**Fig 3. Correlations between dwell-times and attention problems/age.** Scatterplots showing bivariate correlations between dwell-times and attention problems/age.

were unable to significantly predict either measure (RT-variability: $F(3, 108) = .348$, $R^2 = .009$, $p = .790$; Task-performance: $F(3, 108) = 2.02$, $R^2 = .051$, $p = .124$). Similar null-findings were observed for dwell-times from our HMM-PLI model (RT-variability: $F(3, 108) = 1.58$, $R^2 = .042$, $p = .198$; Task-performance: $F(3, 108) = .467$, $R^2 = .012$, $p = .705$).

However, dwell-times from our HMM-Power model were supportive of our primary hypothesis: they significantly predicted reaction-time variability ($F(3, 108) = 3.64$, $R^2 = .092$, $p = .015$) and task-performance ($F(3, 108) = 3.68$, $R^2 = .092$, $p = .014$). These effects were more prominent in those with ADHD compared to controls, for both RT-variability: (ADHD: $F(3, 52) = 3.09$, $R^2 = .151$, $p = .035$; Controls: $F(3, 52) = .69$, $R^2 = .038$, $p =. 56$), and task-performance (ADHD: $F(3, 52) = 2.57$, $R^2 = .129$, $p = .064$); Controls: $F(3, 52) = .843$, $R^2 = .046$, $p = .48$).

To better understand these significant effects, we examined the bivariate correlations between dwell-times in each state of the HMM-Power model and reaction-time variability/ task-performance. Dwell-time in state 2 (characterized by relatively high Alpha/Beta and relatively low Delta/Theta) was negatively correlated with reaction time variability ($r = -.30$, $p = .001$) and positively correlated with task-performance ($r = .22$, $p = .021$); dwell-time in state 4 (characterized by relatively low power across all frequency-bands; hypo-activity), was positively correlated with reaction-time variability ($r = .30$, $p = .002$). Scatterplots for results across all participants are presented in Fig 4; scatterplots comparing ADHD versus controls are presented in Fig 5.

**3.3.2 Secondary hypotheses.** We then tested our secondary hypothesis: that the relationship between cortex-wide oscillatory dynamics and response control would be moderated by

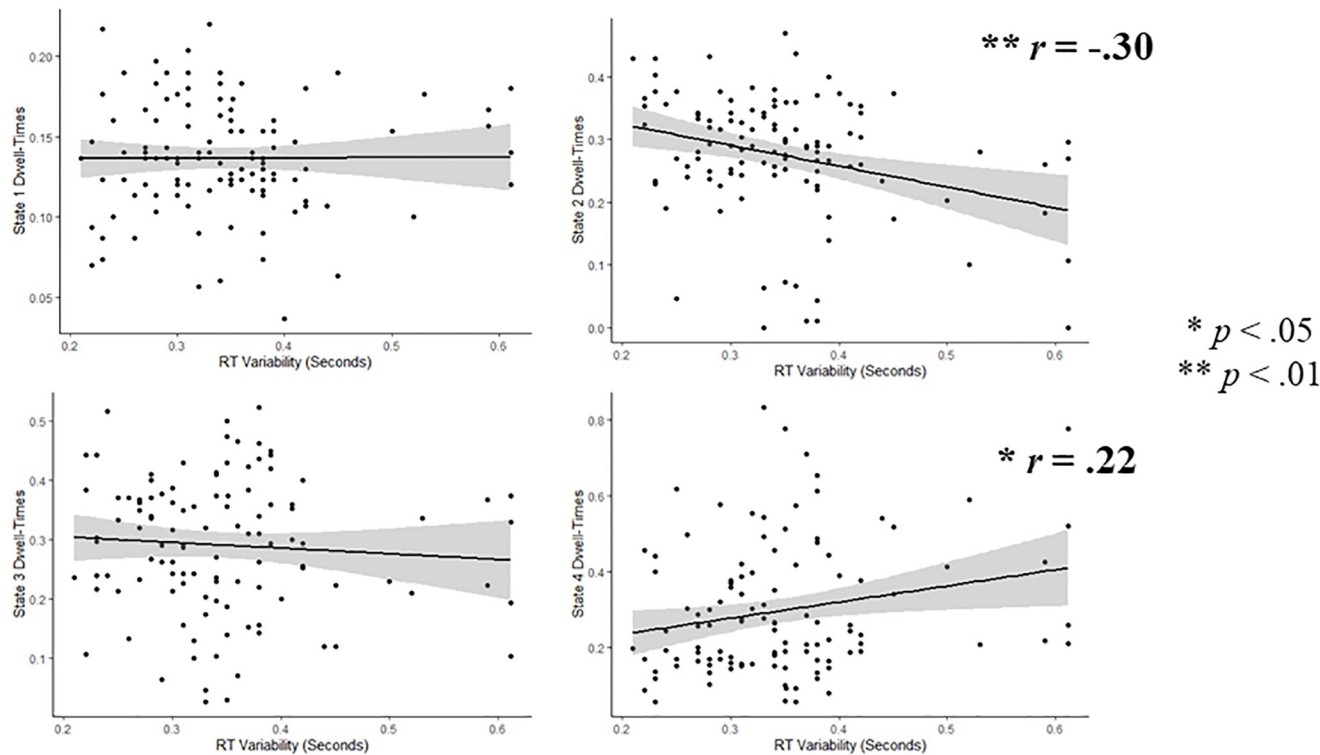

**Fig 4. Correlations between HMM-power dwell-times and RT-variability.** Scatterplots showing bivariate correlations between dwell-times and RT-variability.

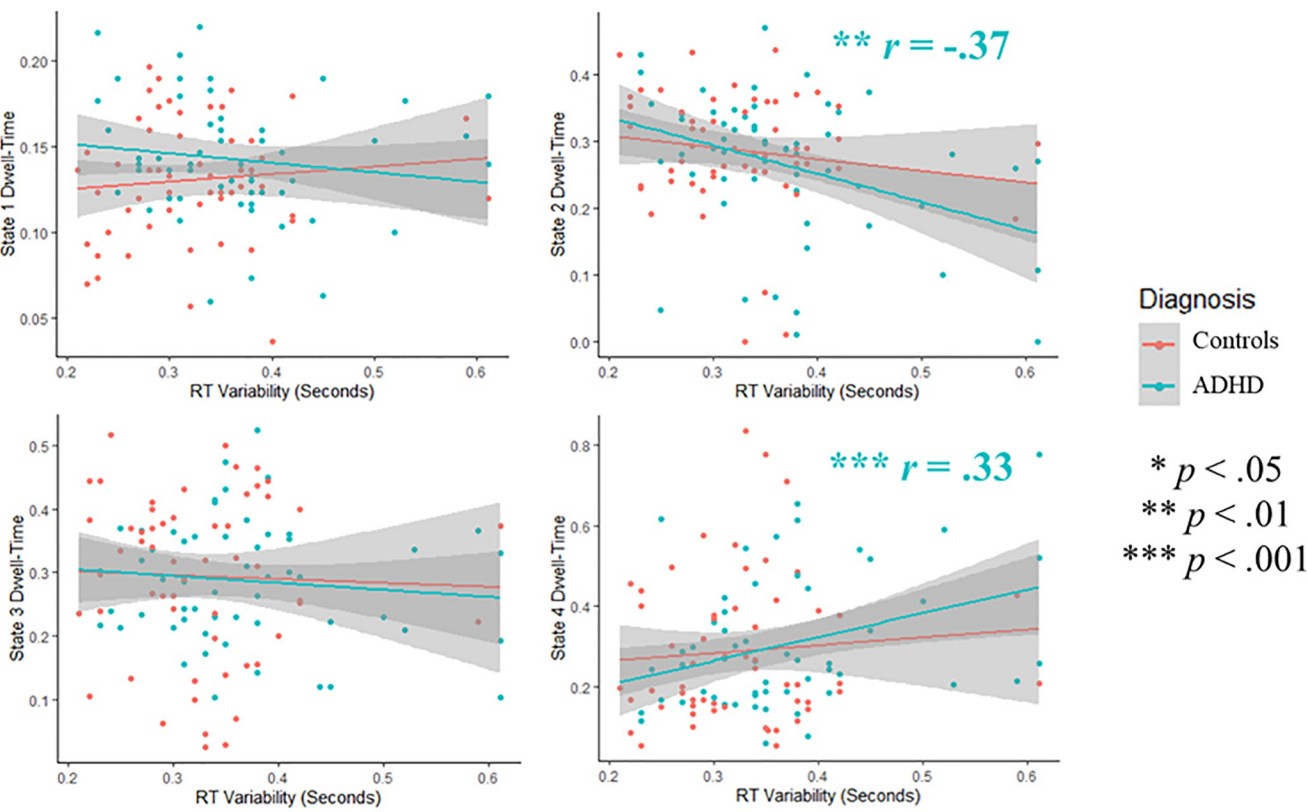

**Fig 5. Group differences in the correlation between HMM-power dwell-times and RT-variability.** Scatterplots showing bivariate correlations between dwell-times and RT-variability, for both controls and ADHD.

sex. For the regression model using HMM-Power dwell-times to predict perceptual decision-making abilities, we found a marginally significant moderating effect of sex for reaction-time variability ($p = .064$). Close inspection of this effect revealed that dwell-time in state 2 was associated with lower reaction-time variability in males ($r = -.429$, $p < .001$) compared to females ($r = -.091$, p = .501; see Fig 6). Fisher's $r$-to-$z$ transform revealed this to be a marginally significant difference ($p = .059$).

**3.3.3 Exploratory analyses: ADHD and Sex.** Using the same procedure (*F*-test for the significance of a subset of variables), we then tested whether the inclusion of both sex and ADHD interaction terms provided a significantly better fit to the model (Reduced model: dwell-times, dummy-coded sex variable, dummy-coded ADHD variable; Full model: additional inclusion of multiplicative terms between: dwell-times and sex, dwell-times and ADHD, and dwell-times, sex, and ADHD). This allowed us to assess whether the sex differences observed in our secondary hypothesis were dependent on ADHD-status. This failed to reach significance ($p = .398$), suggesting the effect of sex is not moderated by ADHD diagnosis (rather, it is present in both ADHD and controls).

Then, in a more focused set of analyses with similar purpose, we directly tested the strength of the result from our primary hypothesis–that dwell-times in states 2 and 4 were associated with response control–in ADHD-males, ADHD-females, Control-males, and Control-females. The strongest (and only significant) relationship between cortical dynamics (HMM-power dwell-times in states 2 and 4) and response control existed in ADHD-males (RT-variability: *F*

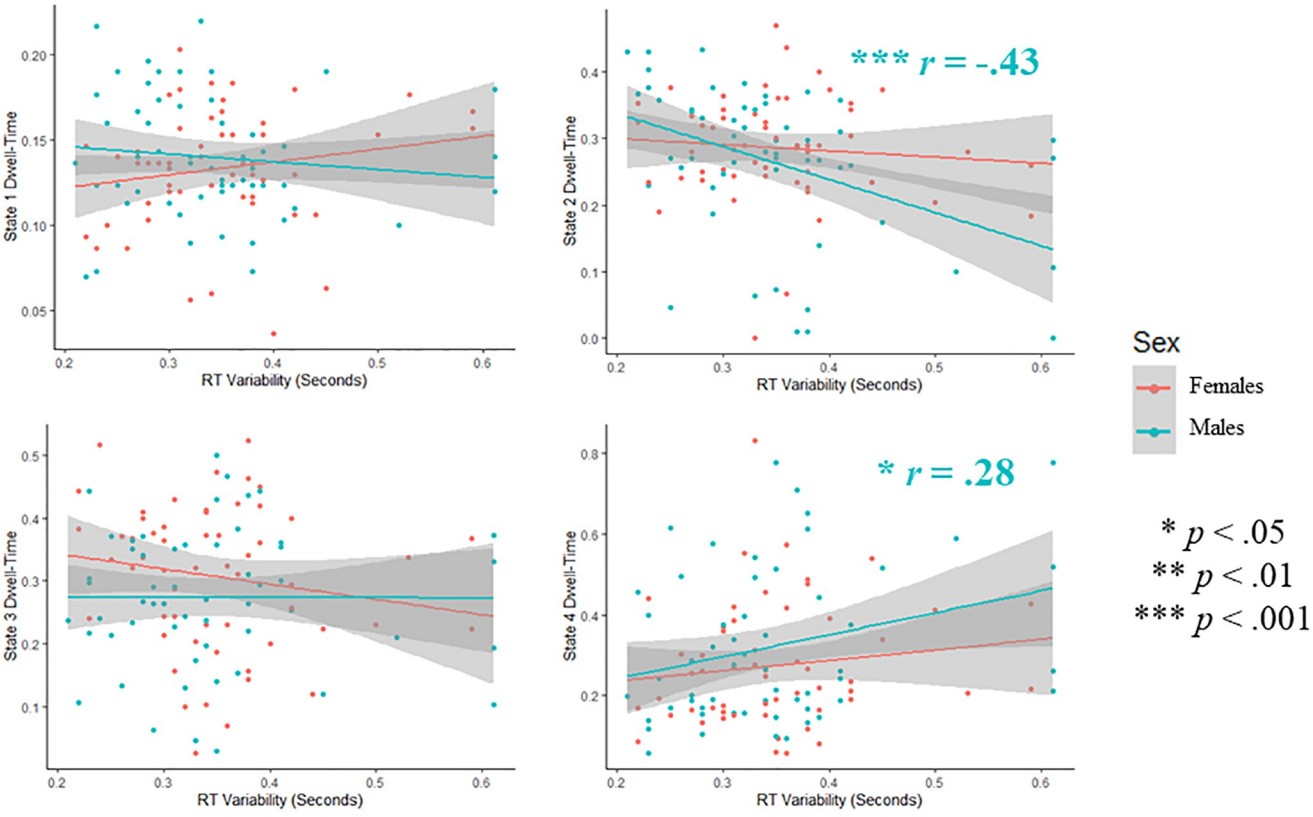

**Fig 6. Sex differences in the correlation between HMM-power dwell-times and RT-variability.** Scatterplots showing bivariate correlations between dwell-times and RT-variability, for both males and females.

$(2, 25) = 4.10$, $R^2 = .247$, $p = .029$). This was followed by very weak/non-significant relationships in control-males (RT-variability: $F(2, 25) = 1.30$, $R^2 = .093$, $p = .291$), ADHD-females ($F(2, 25) = .490$, $R^2 = .034$, $p = .684$), and Control-females ($F(2, 25) = .148$, $R^2 = .012$, $p = .863$).

To summarize, our initial hypothesis test demonstrated that greater reaction-time variability (lower response control) was associated with shorter dwell-times in state 2 (low delta/theta, high alpha/beta) and longer dwell-times in state 4 (hypo-activity across frequency bands). These exploratory analyses clarified the effect, demonstrating that the relationship is only present in males with ADHD; not ADHD-females, control-females, or control-males.

### 3.4 Sensitivity analyses: Examining models with a variable number of states

Unsupervised clustering techniques in general, and hidden Markov models specifically, require one to select the number of states/clusters a priori. To ensure that our results were not too closely dependent on this somewhat arbitrary selection, we tested whether consistent results were observed across a range of models with a varying number of states, as is best practice [56]. Specifically, we re-ran our validation analyses and hypothesis tests for HMM-Power models created with 3, 5, and 6 states.

Each model passed our validation analysis: dwell-times significantly differed between EO and EC-rest (see S1 Table). For hypothesis testing, marginally significant support for our primary hypothesis was found in the 5-state (RT-variability: $p = .086$; Task-performance: $p = .069$) and 6-state models (RT-variability: $p = .059$; Task-performance: $p = .045$). No support

was found in the 3-state model. For our secondary hypothesis (moderating effect of sex), a marginally significant effect was found in our 6-state model ($p = .064$), but not the 3 or 5 state models.

In the 6-state model, the moderating effect of sex was in line with that observed in the 4-state model: dwell-time in state 4 ($\mu$ parameters: Delta = -.60, Theta = -.41, Alpha = .79, Beta = .19) was correlated with reaction-time variability in males ($r = -.45$, $p < .001$), but not females ($r = -.049$, $p = .72$). This was a significant difference (Fisher's $r$-to-$z$ transform: $p = .027$).

Taken together, the 3-state model appeared to distinguish between EO and EC rest, yet this information was unrelated to response control, age, and behavioural measures. The 5 and 6-state models provided marginal support for our primary hypothesis, and the 6-state model provided significant and marginally significant support for both our primary and secondary hypotheses (respectively) that was qualitatively similar to our 4-state model.

## 4. Discussion

In the current study, we first replicated previous findings by demonstrating that reaction-time variability and commission error-rate on a simple perceptual decision-making task are significantly higher in children and adolescents diagnosed with ADHD compared to age-matched controls without ADHD [5, 6]. These findings provide further support for computational models of reaction-time variability in ADHD implicating inefficient 'evidence accumulation' [4]. We then extended previous findings by investigating the specific neural mechanisms that help explain this reduced response control. To do so, we examined whether dynamic changes in three distinct electrophysiological phenomena (phase-synchrony, modularity, oscillatory power) during the resting-state were associated with response control on a separate simple perceptual decision-making task in both the children with and without ADHD. We took phase-synchrony to index the amount of communication between neuronal populations [29, 55], modularity to index the extent to which this communication forms a network with a segregated topology [60], and oscillatory power to index the local synchrony of neuronal populations within these networks [61]. Finally, we tested whether these mechanisms are similarly dysregulated in males and females, with and without ADHD.

### 4.1 Electrophysiological mechanisms underlying differences in response control

Overall, we found that dynamic changes in the *power* of cortical network dynamics, as opposed to changes in their modularity or phase-synchrony, were most associated with differences in response control. Results suggested that, in those with ADHD, response control (reaction-time variability and task-performance) was supported by a neural state with relatively high alpha/beta power and relatively low delta/theta power (state 2). Considering the extent to which this state was moderated by condition (eyes open versus closed), it may primarily reflect an increase in alpha power (the 'Berger' effect), and is hereafter referred to as the high-alpha state [62–64].

In theory, increases in alpha power support information processing by suppressing task-irrelevant stimuli [62]. Such accounts are grounded in information-theory: high alpha power reflects increased dependence among a population of neurons, which necessarily reduces the amount of information that can be carried in the population's firing pattern (information through desynchronization hypothesis) [34]. Increases and decreases in alpha power are primarily modulated by the frontoparietal network (FPN; encompassing regions in dorsolateral prefrontal and posterior parietal cortex), with subcortical support from the thalamus [65].

Through this lens, our results suggest that the mechanisms used by the FPN to initiate a high alpha-power state that suppresses irrelevant processing help to attenuate response control deficits in those with ADHD. In controls, however, these mechanisms seem to have less bearing on response control.

While implicating FPN function is speculative, it accounts for several lines of extant evidence. First, mechanisms mediated by the FPN are thought to directly support evidence accumulation, providing a plausible explanation for why their initiation might support response control. In fMRI, FPN activation covaries with the presentation of decision-relevant sensory information [35, 66]. Moreover, in diffusion-weighted MRI, fractional anisotropy of the superior longitudinal fasciculus (large white matter tract connecting dorsolateral prefrontal cortex with posterior parietal cortex) is associated with the rate of evidence accumulation, as estimated from the centroparietal positivity (a component of the event-related potential that exhibits all the classical properties of an evidence accumulator) [14, 67]. Second, Cai et al. (2018) [7], who investigated the task-related dynamics in fMRI associated with response control in ADHD, found that functional connectivity between the FPN and salience network is associated with evidence accumulation (estimated using drift-diffusion modeling).

Counterintuitively, dwell-times in the high-alpha state, which were most associated with response control, did not significantly differ between ADHD and controls. Instead, the same neural state benefitted response control to a greater extent in those with ADHD compared to controls. Interestingly, this is similar to research by Duffy et al. (2021) [8], who found that integration (increased participation coefficient) between both the DMN and SAL, as well as DMN and sensorimotor network, are associated with response control, yet do not differ between ADHD and controls. One explanation as to why we observed this 'mismatch' in the current study is that disruptive processing which needs to be suppressed in ADHD (through alpha gating mediated by the FPN, and thus dwell-time in the high-alpha state) is simply not present in controls. If so, suppression of this processing would support response control in ADHD yet have little benefit to controls. Such an interpretation suggests that these mechanisms are not *dysregulated* in ADHD, per se, but that regulation of these mechanisms impacts response control differently in ADHD compared to controls.

Interestingly, we found that the dynamics associated with response control were distinct from the dynamics associated with self-reported measures of inattention. Indeed, the association between reaction-time variability and dwell-times in the high-alpha state and hypo-activation state were significant after controlling for CBCL-Inattention scores. This is similar to fMRI research by Cai et al. (2018) [7], who found dissociable network dynamics underlying reaction-time variability and inattention (SAL–FPN connectivity predicted reaction-time variability, while SAL–DMN connectivity predicted inattention). Our results, then, appear to provide converging evidence for this finding from a distinct modality.

## 4.2 Sex differences

Typically, females with ADHD present with distinct phenotypes compared to males with ADHD: lower hyperactivity, externalizing behaviours, as well as greater intellectual difficulties, depression and anxiety [68]. This complicates research investigating the neural mechanisms that underlie ADHD-symptoms, as it is often unclear the extent to which sex differences observed in neural activity are simply a result of phenotypic heterogeneity. To account for this, we purposely created a homogenous sample, where males and females did not significantly differ by age, ADHD-relevant behaviours or anxiety. Controlling for behavioural differences in this way increases our confidence that the observed sex differences in the brain are not a result

of confounding differences in behaviour, but rather, reflect differences in the brain-behaviour relationship.

Within this relatively homogenous sample, we observed that dwell-time in the high-alpha state (reflecting regulation of oscillatory power by the FPN, putatively) supports response control in males, but has less of a bearing on response control in females (a marginally significant difference). This would imply that females rely on alternative mechanisms for attenuating response control deficits than males; mechanisms which were not captured in the current analyses. It would also imply that previous research linking cortical-network activity with response control in ADHD may be driven by an overrepresentation of males [8, 11, 19].

We expect these findings to inspire further research into the neural mechanisms that help attenuate response control deficits in females. Some insight may be gleaned from research by Murray et al. (2021) [69]: while males tend to dwell in states with integration between the default mode and salience networks, females tend to dwell in states with integration between the default mode and dorsal attention network (DAN). Interestingly, functional connectivity between the DMN and SAL network is associated with reduced response control [8]. This would implicate DMN-DAN integration as a potential compensatory mechanism evoked by females.

## 5. Conclusions

In this study, we investigated the relationship between response control (reaction-time variability, task-performance) on a simple perceptual decision making task and resting-state cortical network dynamics in those diagnosed with ADHD and a set of typically-developing controls. We focused on a subset of network properties thought to be particularly relevant to ADHD (phase-synchrony, modularity, oscillatory power) and tested whether differences in these properties help explain differences in response control. Based on the consistent lack of segregation between large-scale functional networks in ADHD (default-mode interference), we hypothesized that exhibiting an electrophysiological profile with high modularity would be associated with reductions in response control. Yet, contrary to this hypothesis, we found that differences in response control were most associated with changes in oscillatory power. Moreover, this relationship was predominantly driven by males, and relatively weak in females. Based on the characteristics of the observed EEG dynamics (high alpha, modulated by eyes-open/eyes-closed status), we suggest that alpha-suppression mechanisms may help those with ADHD–particularly males–attenuate processing that is disruptive to response control.

## Supporting information

**S1 Table. Validation of HMM-power models across a variable number of states.** (XLSX)

## Author Contributions

**Conceptualization:** Jonah Kember, Erin Panda, Ayda Tekok-Kilic.

**Data curation:** Jonah Kember.

**Formal analysis:** Jonah Kember.

**Investigation:** Jonah Kember.

**Methodology:** Jonah Kember.

**Supervision:** Ayda Tekok-Kilic.

**Validation:** Jonah Kember.

**Visualization:** Jonah Kember.

**Writing – original draft:** Jonah Kember, Lauren Stepien, Ayda Tekok-Kilic.

**Writing – review & editing:** Lauren Stepien, Erin Panda, Ayda Tekok-Kilic.

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
