## [Decision Letter · Decision Letter 0]

13 Feb 2023

PONE-D-22-29382Resting-state EEG dynamics help explain differences in response control in ADHD: Insight into electrophysiological mechanisms and sex differencesPLOS ONE

Dear Dr. Kember,

Thank you for submitting your manuscript to PLOS ONE. After careful consideration, we feel that it has merit but does not fully meet PLOS ONE’s publication criteria as it currently stands. Therefore, we invite you to submit a revised version of the manuscript that addresses the points raised during the review process. I encourage the authors to thoroughly address the issues highlighted by reviewers, keeping in special account the remarks from reviewer #2. While the manuscript is interesting, it is not always easy to follow the authors reasoning regard e.g. which is the aim and the meaning of a specific analisys and so on.

We look forward to receiving your revised manuscript.

Kind regards,

Federico Giove, PhD

Academic Editor

PLOS ONE

Journal Requirements:

“The study was partially supported by Brock University Council for Research in Social Sciences (CRISS), awarded to ATK and EP.

Reviewers' comments:

Reviewer's Responses to Questions

**Comments to the Author**

1. Is the manuscript technically sound, and do the data support the conclusions?

Reviewer #1: Partly

Reviewer #2: Partly

2. Has the statistical analysis been performed appropriately and rigorously? 

Reviewer #1: I Don't Know

Reviewer #2: I Don't Know

3. Have the authors made all data underlying the findings in their manuscript fully available?

Reviewer #1: Yes

Reviewer #2: Yes

4. Is the manuscript presented in an intelligible fashion and written in standard English?

Reviewer #1: Yes

Reviewer #2: Yes

5. Review Comments to the Author

Reviewer #1: Statistical p-values should be given with * referring p<0.05 and ** referring p<<0.05 in tables.

The authors should discuss the findings in recent studies including graph theoretical connectivity analysis in ADHD, doi: 10.1016/j.bspc.2022.103626.

How did the authors compare the groups ? what are the features (i.e. indicators) ?

Did they check the reliability of the results by examining several methods different from each other.

Reviewer #2: Overall comments:

An interesting study aiming to pinpoint the impact of dysregulated cortical network / response-control pathway in individuals with Attention Deficit Hyperactivity Disorder. The study benefits from a good sample size, using psychometrically validated scales, and the analyses appear thorough.

My biggest issue is that many readers may not be familiar with the Hidden Markov Modelling statistical approach. The way the results are presented and discussed rarely refer to EEG frequency bands that define each of the states the authors identified in their analyses. On the limited occasions that they do so (e.g. p22 where state 2 defined as relatively high alpha/beta and relatively low delta/theta), it enables the reader to contextualise the findings with regard to functional significance of EEG frequency bands in conjunction with ADHD. However, throughout the majority of the results and discussion it is unclear exactly what the results mean. A further example of this is in section 3.4 (p25) where the authors amend their analysis approach to explore states 3, 5, & 6 (previously it was 4 states). This needs to be clarified throughout; especially as the foreshadowing of the results (page 12) identified that it was the power of the cortical networks that explained the response control differences. This required detail will make the reading of this paper considerably more salient.

Other comments:

Intro: p6/7 - there are no hypotheses for sex differences - do the authors have a priori direction for these? This also ties in with issues regarding structure of the introduction which deviates from standard formatting of a research article.

Method: p8 – the grouping and matching (lines 1-4) are unclear, as it reads that females were matched by age, and males were matched by subtype (not specified). Please reword accordingly.

p9 Section 2.3: CBCL – As it has been stated only 3 subscales were included, please clarify number of items used in the study and provide the Cronbach’s alpha.

C3-SR-S / SCARED– please provide the Cronbach’s alpha values observed here.

Results:

Please state which programme was used to analyse the data.

The behavioural results in sections 3.1 and 3.2 are missing the inferential statistics, and have p values only reported, whereas inferential statistics have been included in section 3.2.2. Please be consistent throughout the results; another example is p15 ‘As shown in Table 2…’ three non-significant findings are referred to in text, but only one p value reported. Please also consistently state the direction of the result.

There are inconsistencies in statistical reporting also. It would be clearer to the reader to adhere to one format (p values) when referring to significance / non-significance rather than the variation presented in Tables 2 and 4. For example, Table 4 also includes a significant p value (state 3 power). If you have a reason to use this alternative format for significance reporting then please can you explain why.

There are also inconsistencies in the body of the text in how the p value is reported for significant findings when the p value is less than .001. In some cases, p<.001 has be used, other times an exact p value is reported e.g. p = .0009 (p23).

The table legends do not follow standard formatting, where the legend should be above the table. Similarly, the legends for the figures should be below the figure. Much text below the figures should actually be incorporated in the body of the text. On some occasions it has been included in text also, therefore there is repetition.

p18: Where several non-significant findings are combined in one statement, provide minimum p value i.e. p≥ as opposed to p>.

p20: HMM-Power dwell-times: age negatively correlated with dwell time in state 3 but coefficient is positive. Text and Figure 3 do not correspond. The legend requires rewording as age is not a phenotype.

p22: Figure 4 correlations – remove *** p<.001 as there are no results at this level of significance presented

p24: Figure 6 – text states increased dwell-times with increased state 2 – but coefficient is a negative correlation. Reword the females result to state non-significant rather than attenuated.

P25 – ‘in the four groups created by our two variables of interest’ - it is not clear what this means.

Please check thoroughly throughout to ensure correct statistical reporting of all the results.

This section would benefit from a summary of the key findings.

Discussion:

p28 – remove the inferential stats. Again, it is not clear what states 2 and 4 relate to therefore not clear how this concurs with Cai et al. (2018).

Please also see my general comments regarding the lack of reference to EEG frequency bands associated with each state. It is hard for the lay reader to follow how the findings fit with established literature.

Minor comments

Intro:

Acronyms need to be written in full initially e.g. p4 line 4 states ‘dysregulated FPN – SAL interactions’, but the acronym for FPN is not stated till p33, I cannot find a full reference for SAL in the manuscript.

p5 reference Stam et a – missing the ‘l’

Method:

P7: Research ethics board – xxxx – is this a typo? It is not a blinded submission

Inconsistencies in reporting the ages of participants. The methods state ages 8-18 were targeted, the abstract states 8-16, the text in the results refers to 16.83. Once the demographic information in the results has been moved to the Method please be consistent regarding the age range.

p8 lines 5-12 - signpost reader to section 2.3 regarding these materials.

p11 - below modularity equation. ‘Where □…’ – typo

Section 2.3: Connors – missing closing bracket (positive and negative impression

Please proof read thoroughly.

6. PLOS authors have the option to publish the peer review history of their article (what does this mean?). If published, this will include your full peer review and any attached files.

Reviewer #1: No

Reviewer #2: No

---

## [Author Response · Author response to Decision Letter 0]

30 Mar 2023

Please see: 'Response to Reviewers.docx'

---

## [Decision Letter · Decision Letter 1]

15 May 2023

Resting-state EEG dynamics help explain differences in response control in ADHD: Insight into electrophysiological mechanisms and sex differences

PONE-D-22-29382R1

Dear Dr. Kember,

We’re pleased to inform you that your manuscript has been judged scientifically suitable for publication and will be formally accepted for publication once it meets all outstanding technical requirements.

Kind regards,

Federico Giove, PhD

Academic Editor

PLOS ONE

Additional Editor Comments (optional):

Reviewers' comments:

Reviewer's Responses to Questions

**Comments to the Author**

1. If the authors have adequately addressed your comments raised in a previous round of review and you feel that this manuscript is now acceptable for publication, you may indicate that here to bypass the “Comments to the Author” section, enter your conflict of interest statement in the “Confidential to Editor” section, and submit your "Accept" recommendation.

Reviewer #2: All comments have been addressed

2. Is the manuscript technically sound, and do the data support the conclusions?

Reviewer #2: (No Response)

3. Has the statistical analysis been performed appropriately and rigorously? 

Reviewer #2: (No Response)

4. Have the authors made all data underlying the findings in their manuscript fully available?

Reviewer #2: (No Response)

5. Is the manuscript presented in an intelligible fashion and written in standard English?

Reviewer #2: (No Response)

6. Review Comments to the Author

Reviewer #2: (No Response)

7. PLOS authors have the option to publish the peer review history of their article (what does this mean?). If published, this will include your full peer review and any attached files.

Reviewer #2: No

---

## [Editor Report · Acceptance letter]

22 May 2023

PONE-D-22-29382R1 

Resting-state EEG dynamics help explain differences in response control in ADHD: Insight into electrophysiological mechanisms and sex differences 

Dear Dr. Kember:

I'm pleased to inform you that your manuscript has been deemed suitable for publication in PLOS ONE. Congratulations! Your manuscript is now with our production department. 

Kind regards, 

on behalf of

Dr. Federico Giove 

Academic Editor

PLOS ONE